# Protease-activated receptor 2 deficient mice develop less angiotensin II induced left ventricular hypertrophy but more cardiac fibrosis

Albrecht Meyer zu Schwabedissen[1], Silvia Vergarajauregui[2], Marko Bertog[3], Kerstin Amann[1], Felix B. Engel[2], Christoph Daniel[1]*

1 Department of Nephropathology, Universitätsklinikum Erlangen, Friedrich-Alexander University Erlangen-Nürnberg (FAU), Erlangen, Germany, 2 Department of Nephropathology, Institute of Pathology and Department of Cardiology, Experimental Renal and Cardiovascular Research, Friedrich-Alexander-Universität Erlangen-Nürnberg (FAU), Erlangen, Germany, 3 Institute of Cellular and Molecular Physiology, Friedrich-Alexander University Erlangen-Nürnberg (FAU), Erlangen, Germany

* Christoph.Daniel@uk-erlangen.de

## Abstract

### Aims

Activation of Protease Activated Receptor 2 (PAR2) has been shown to be involved in regulation of injury-related processes including inflammation, fibrosis and hypertrophy. In this study we will investigate the role of PAR2 in cardiac injury in a mouse model of hypertension using continuous infusion with angiotensin II.

### Methods

Hypertension was induced in 12 weeks old wildtype (wt, n = 8) and PAR2 deficient mice (n = 9) by continuous infusion with angiotensin II for 4 weeks using osmotic minipumps. At the end, hearts were collected for analysis of left ventricular hypertrophy (LVH), myocardial capillary supply, fibrosis and localization of PAR2 expression using histological, immunohistological and mRNA expression analysis techniques. In addition, rat cardiac fibroblasts were treated with angiotensin II and PAR2 was inhibited by a blocking antibody and the PAR2 inhibitor AZ3451.

### Results

Cardiac PAR2 mRNA expression was downregulated by 40±20% in wt mice treated with AngII compared to untreated controls. Four weeks after AngII treatment, LVH was significantly increased in AngII-treated wt mice compared to similarly treated PAR2-deficient animals as determined by relative heart weight, left ventricular cross-sectional area, and analysis of ventricular lumen area determined on sections. Treatment of wt mice resulted in an approximately 3-fold increase in cardiac expression of FGF23, which was 50% lower in PAR2-deficient animals compared to wt animals and therefore no longer significantly different from expression levels in untreated control mice. In contrast, cardiac interstitial fibrosis

**Data Availability Statement:** The source data underlying this article are shared on: DOI: 10.6084/m9.figshare.26430022.

**Funding:** This work presented in this manuscript was supported by ELAN-Fonds of the Friedrich-Alexander University Erlangen-Nürnberg for the project "PAR-2 in heart and kidney," awarded to CD, and by Deutsche Herzstiftung e.V., awarded to FE. The funders had no role in study design, data collection and analysis, decision to publish, or preparation of the manuscript.

**Competing interests:** NO authors have competing interests.

**Abbreviations: PAR2**, Protease activated receptor 2, belongs to group of four G-Protease-coupled receptors that are activated by proteolytic activity of numerous enzymes; **AngII**, Angiotensin II, Peptide hormone belonging to the Renin-Angiotensin-Aldosteron system; **Factor Xa**, serine protease, capable of activating PAR2; **TGF-β**, Transforming growth Factor β, Cytokine, most important pro-fibrotic cytokine with further effects on hypertrophy and inflammation; **P-ERK1/2**, phosphorylated extracellular-signal regulated kinases 1/2, serine/threonine kinases that belong to the group of mitogen-activated protein kinases (MAP kinases); **Caveolin-1**, Protein found on calveolae of various cell types that causes the internalisation of receptors. High concentrations of calveolin-1 are found in fibrocytes and endothelial cells.

was significantly higher in PAR2-deficient mice compared to similar treated wt controls, as assessed by Sirius Red staining (>3-fold) and collagen IV staining (>2-fold). Additional experiments with isolated cardiac fibroblasts showed induction of pro-fibrotic genes when treated with PAR2 inhibitors.

## Conclusion

In angiotensin II-induced cardiac injury, PAR2 deficiency has an ambivalent effect, enhancing fibrosis on the one hand, but reducing LVH on the other.

## 1. Introduction

Approximately 31% of the world's adult population suffers from hypertension attributing to millions of premature deaths worldwide [1]. Hypertension is also one of the major risk factors for left ventricular hypertrophy (LVH), myocardial infarction, ischemic or hemorrhagic stroke, and cardiac remodelling [2]. The latter is defined as the pathological transformation of the heart into tissue with increased connective tissue content and hypertrophy of cardiac myocytes [3]. Cardiac remodelling is triggered and influenced by a variety of factors via receptors, including one of the most potent vasoconstrictors angiotensin II (AngII) but also by protease-activated receptors (PAR) [4,5]. PARs are a group of four (PAR1, PAR2, PAR3, PAR4) G protein-coupled receptors [6]. The PAR2 is located on the cell surface [6] and is expressed in various tissues and cells, particularly in smooth muscle cells, endothelial cells and immune cells [7–9]. The receptor is activated by serine proteases such as coagulation factors VIIa and Xa, trypsin and various human kallicreins and tryptase, which cleave the extracellular N-terminal domain of the receptor to form a bound ligand, which then binds to the second extracellular membrane-spanning loop [10]. The intracellular signalling pathways of the PAR2 are complex and dependent on cell type, interaction with other receptors, and the concentration and specific cleavage site of activating or inhibiting proteases [11,12]. Hereby PAR2-receptor is involved in regulation of inflammation, neovascularization and fibrosis [13–15]. Interestingly PAR2 receptor can exert both pro-fibrotic [16,17] and anti-fibrotic [13] effects. Furthermore, PAR2 is involved in regulating cardiac hypertrophy via phosphorylation of ERK1/2 and p38 [18].

In this study, we investigated the role of PAR2 in the development of LVH, cardiac vascular wall thickening as well as cardiac fibrosis after AngII-treatment using PAR2 deficient mice.

## 2. Material and methods

### 2.1 Animals

All experiments were carried out using wild type and PAR2-knock out (PAR2$^{-/-}$; B6.Cg-F2rl1tmMslb/J) [19] mice (n = 29) with an age of 12-week at the start of the experiment. PAR2$^{-/-}$ mice were obtained from Dr. Martin Steinhoff (Department of Dermatology, University of Münster, Germany). The hypertension model was induced in wildtype (wt, *n* = 8) and PAR2$^{-/-}$ (PAR2 ko, *n* = 9) by subcutaneous implantation of osmotic minipumps (3 μg kg-1 min-1 angiotensin II with Alzet 2004 (ALZET, Cupertino, CA, USA)) for a period of 4 weeks. The osmotic minipumps were filled with angiotensin II dissolved in PBS and then implanted subcutaneously. The mice were anesthetized with isoflurane, analgesia was administered subcutaneously with buprenorphine (0.05mg/kg bw) before the procedure, the fur was shaved

dorsally on the right flank, the skin was disinfected, the skin was incised about 1 cm long and a subcutaneous pocket was prepared with blunt scissors. The filled osmotic minipump was then placed in the prepared skin pocket and the skin closed with single-button sutures. On the following day, analgesia was again administered subcutaneously with buprenorphine (0.05mg/kg bw). The pump remained in the animal for 4 weeks until the end of the experiment. Sham operated animals served as controls (wt, $n = 6$; PAR2 ko, $n = 6$). All animals were housed under conditions of 12-h:12-h light–dark cycles and were fed a standard diet with water and food until they were 16 weeks old. All experiments were executed in accordance to the Guide for the Care and Use of Laboratory Animals Directive 2010/63/EU, European Parliament on the protection of animals used for scientific purposes. The experimental protocol for the animal studies was approved by the regional government for the mouse studies ("Regierung von Mittelfranken", Permit number: 54–2532.1-2-53/12) upon recommendation of the local committee for animal care and use. To prevent suffering, the animals were monitored daily after the procedures and twice weekly otherwise, and symptoms such as weight loss of more than 20% or pain on gripping the animals were defined as leading to immediate termination of the experiment. However, no symptoms occurred in this study that would have led to termination. At the end of the experiment, the animals were administered 0.05 mg/kg body weight buprenorphine for analgesia before being placed under general anesthesia with isoflurane. For euthanasia, the mice were exsanguinated after an abdominal incision with blood sampling from the vena cava, followed by median thoracotomy with opening of the great vessels. To ensure good histologic processing, all animals were perfused with 10% dextran (Deltamedica GmbH, Reutlingen, Germany) with 0.2% procaine (Steigerwald Pharma, Darmstadt, Germany) and then with 0.9% NaCl (B. Braun, Melsungen, Germany) before the hearts were removed via the left ventricle. After euthanasia, hearts and kidneys of all animals have been weighted. Hearts were prepared for immunohistochemical analysis and *in-situ*-hybridisation. Fresh frozen samples were collected for real-time PCR.

Extraction of organs and preparation of primary cell cultures were approved by the local Animal Ethics Committee in accordance to governmental and international guidelines on animal experimentation (protocol TS-17/2023 Nephropatho). 3-day-old (P3) Sprague-Dawley rats were decapitated with sterile scissors, and the thorax was opened along the sternum to gain access to the thoracic cavity and remove the heart.

## 2.2 Methods

**2.2.1 Histology and immunohistochemistry.** Hearts of all animals were collected at the end of the experiment and first total heart weight was determined. Then right ventricle was removed and left ventricle was weighted again followed by fixation in 4% formalin buffered in PBS pH 7.6 and processed by dehydration in ascending alcohol series followed by xylol and paraffin embedding. For immunohistochemistry, paraffin sections were deparaffinated and rehydrated using xylol followed by descending alcohol series. Haematoxylin/eosin and Sirius Red stain was performed using standard routine protocols. For immunohistochemistry endogenous peroxidase was blocked by incubation with 3% $H_2O_2$ for 20 min followed by incubation in blocking solution (1% bovine serum albumin (BSA) in Tris(hydroxymethyl)aminomethan-buffered saline supplemented with 0.05% Tween 20 (all purchased from Sigma Aldrich, Deisenhofen, Germany)) for 10 min at room temperature to prevent unspecific binding. The sections were incubated with the following primary antibodies were diluted in blocking solution at 4°C overnight: a rabbit polyclonal antibody directed against collagen I (Biotrend Chemikalien GmbH, Köln, Germany); a goat polyclonal antibody directed against human collagen IV (Southern Biotech, Birmingham, AL, USA), a mouse monoclonal antibody directed against

alpha smooth muscle actin (SMA) (DAKO, Glostrup, Denmark), a rat monoclonal antibody directed against CD31(DAKO, Glostrup, Denmark); a goat polyclonal antibody directed against P-Smad 2/3 (sc-11769; Santa Cruz Biotechnology, Dallas, TX, USA); a rabbit polyclonal antibody directed against p44/42 MAPK (ERK1/2; Thr202/Tyr204) (Cell Signaling Technology Europe B.V., Leiden NL). After washing with Tris-buffer pH 7.6 supplemented with 0.05% Tween 20, sections were incubated for 30 minutes at room temperature with appropriate biotinylated secondary antibodies diluted 1:500 in blocking solution: a horse anti mouse IgG (BA-2001); a rabbit anti-goat IgG (BA-5000); a goat anti-rabbit IgG (BA-1000) (all purchased from Vector laboratories). Bound secondary antibodies were detected using ABC-Kit and DAB-Immpact as a substrate (both from Vector laboratories). Finally, nuclei were counterstained using hemalaun (Merck KGaA, Darmstadt, Germany) and sections were covered with Entellan (Merck KGaA, Darmstadt, Germany) and a coverslip.

**2.2.2 Quantification of Sirius Red stain, immunohistochemistry and grading of histopathological changes.** HE stained heart sections were used to analyze LVH. For this purpose, the total and lumen area of the median sectioned hearts were measured using CellSens software (Olympus Deutschland GmbH, Hamburg Germany). From this, the ratio of total to lumen area was also calculated. In addition, wall thickness of intramyocardial arterioles was measured at 10 different representative sites and averaged. Sirius Red was used for analysis of cardiac fibrosis. Fibrosis was determined using 4 different stainings. Quantification of Sirius Red and α-smooth muscle actin (SMA) stained fibrotic areas, was performed by computer-assisted automatic color recognition using Tissue Studio from Definiens (Definiens, Carlsbad, Califonia, US) with MatLab (The MatchWorks inc., Natick, Massachusetts, US) after the sections were digitized with a Slidescanner (Zeiss Z1, Zeiss, Oberkochen, Germany). Sirius Red stained area was expressed as percentage of the total cardiac tissue section area while the percentage of SMA-positive cells of all cells was determined.

In addition, collagen I and collagen IV-positive areas were scored semi-quantitatively. For each visual field, the degree of fibrosis was scored from 0–4. For this purpose, sections were analyzed with a 10x10 eyepiece grid and the least squares were determined with a positive stain. Fields of view with less than 6 squares were classified as score 0, with 6–10 as score 1, with 11–20 as score 2, with 21–30 squares as score 3 and further scaling was assigned in equal steps. Alpha-SMA was stained to detect myofibroblasts in heart tissue. For evaluation, all preparations were digitised and the percentage of α-SMA-positive cells of the total number of cardiac cells was analysed.

CD31 staining was used to detect cardiac capillary number per cardiomyocytes. For this purpose, the number of cardiomyocytes and capillaries was determined in 10 fields of vision with transversely sectioned cardiomyocytes at a magnification of 400x, followed by calculation of the ratio of capillaries per myocyte.

**2.2.3 Analysis of PAR2 effects using isolated neonatal cardiac rat fibroblasts.** Hearts from 3-day-old postnatal rats were isolated and digested using the gentleMACS Dissociation kit (Miltenyi Biotech GmbH, Bergisch Gladbach, Germany) according to the manufacturer's instructions. For P3 fibroblast enrichment, cells were pre-plated in DMEM-F12/Glutamax TM-I (Life Technologies, Darmstadt, Germany) with 10% fetal bovine serum (FBS, Biowest, Nuaille, France) and penicillin/streptomycin (100 mg/ml) (Life Technologies). After 1 hour, non-adherent cells were discarded, and fibroblasts were cultured in the same medium for 1 day. Cells were then trypsinized and seeded in p12 wells (1 ml) or p24 wells (0.5 ml) at a density of $2x10^5$ cells/ml. After 24 hours, cells were starved in medium containing 0.1% FBS for 24 hours before stimulation with Ang II (1 µM), in the absence or presence of PAR2 inhibitors. PAR2 was inhibited with either a mouse anti-PAR2 monoclonal antibody (1 µg/ml, clone SAM11, Santa Cruz Biotechnology Inc, Santa Cruz, CA, USA) [20] or the PAR2 inhibitor

AZ3451 [21] (2 μM, Tocris Bioscience, Bristol, UK). Controls either remained untreated or received an isotype control antibody (1 μg/ml mouse IgG2a, 10-458-C025, Exbio Antibodies, Vestec, Czech Republic). Cells were then harvested and RNA isolated for expression analysis. 3–5 replicates were performed per condition. For immunofluorescence staining, fibroblasts were fixed with 4% formalin for 15 min and permeabilized with ice-cold methanol at 4°C for 10 min after washing in PBS. Fibroblasts were then incubated for 1 hour at 37°C with rabbit monoclonal anti-P-ERK1/2 (4370L, Cell signalling Technology Europe BV, Leiden, The Netherlands) or rabbit polyclonal anti-P-Smad3 (9520, Cell signalling Technology Europe BV) diluted in 1% BSA in PBS pH 7.6. After washing with PBS pH 7.6, the coverslips were incubated with donkey anti-rabbit IgG Alexa Fluor 568 (Life Technologies GmbH, Darmstadt, Germany) diluted 1:500 in 1% BSA in PBS pH 7.6. After another washing step, the cells were coverslipped with TrueView DAPI (Vector Laboratories Inc.). Finally, 5 images per preparation were captured with a 20x objective on a confocal laser scanning microscope (Zeiss 710, Zeiss, Oberkochen, Germany) and the percentage of positive cells was determined using QuPath software version 0.2.3 [22]. Detailed parameters used for evaluation of P-ERK1/2 and P-Smad3-positive cardiac rat fibroblasts are listed in S1 File. Since numbers of positive cells varied from experiment to experiment, these data were normalized to the level of untreated controls.

**2.2.4 Quantitative real-time PCR and *in situ* hybrization.** We analysed levels of RNA expression of various activators of PAR2 as well profibrotic proteins. Real time PCR of PAR2 was also performed to prove effectiveness of knockout procedure. For expression analysis, RNA was isolated from fresh frozen heart or isolated cardiac fibroblasts using the RNeasy Fibrous tissue mini Kit (Qiagen, Venlo, Netherlands). First-strand cDNA was synthesized with TaqMan reverse transcription reagents (Applied Biosystems, Darmstadt, Germany) using random hexamers as primers. Reactions without multiscribe reverse transcriptase were used as negative controls for genomic DNA contamination. PCR was performed with a Step One Plus Sequence Detector System FastSYBR Green Universal PCR Master Mix (Applied Biosystems), as described previously [23]. All samples were run in triplicate. Specific mRNA levels in mice were calculated and normalized to 18S (fw primer: `TTGATTAAGTCCCTGCCCTTTGT`; rv primer: `CGATCCGAGGGCCTCACTA`) as housekeeping gene and shown as fold change expression levels using the ΔΔct method. The Primers used for expression analyses of mouse collagen I, FGF23, cardiac troponin, BNP and PAR2 as well as rat collagen 1, collagen 4, SMA, fibronectin and TGF-ß are shown in S1 Table.

Formalin-fixed paraffin-embedded sections of 4μm thickness were stained with RNA-Scope method to get information about the localization as well as the quantity of the PAR2-RNA expression using the PAR2-specific probe Mm-F2rl1 (417541; ACD, Hayward, CA, USA). The hybridization protocol was performed as described previously [24].

## 2.3. Statistical methods

Normal distribution was tested using the Kolmogorov-Smirnov test. The variance homogeneity was evaluated using the Brown-Forthsythe test. In the case of normally distributed samples and homogeneity of variance, an ANOVA with Bonferroni multiple comparison test was performed. When samples were not normally distributed and for all experiments *in vitro* due to low numbers of replicates (n = 3–5), the Kruskal-Wallis test was used as a nonparametric alternative, followed by a Dunn posthoc test. Values were excluded as outliers if they deviated from the mean by more than two standard deviations. All calculations were performed with GraphPad Prism version 9.0 and data were shown as box-plots with a box showing the 95% confidence interval and whiskers indicating minimum and maximum values. p-values < 0.05 were

assumed to be significant different and marked by asterisks (* = $p < 0.05$, ** = $p < 0.01$, *** = $p < 0.001$; **** = $p < 0.0001$).

## 3. Results

Based on previous knowledge of the function of the PAR2 receptor, we investigated the role of the PAR2 receptor in cardiac fibrosis, remodelling, as well as smooth muscle and left ventricular hypertrophy.

### 3.1 PAR2 deficiency reduced LVH in the AngII mouse model

To assess myocardial hypertrophy, relative heart weight to body weight, left ventricular cross-sectional area, left ventricular cross-sectional lumen, and FGF23 gene expression were determined (Fig 1). Relative heart weights showed no significant differences in the control groups of both genotypes, with mean relative heart weights between 0.54% and 0.57% of total body weight (Fig 1A). In contrast, the relative heart weight of AngII-treated wt mice was significantly increased by at least 12% compared to all other groups. In particular, the mean relative heart weight in the AngII-treated PAR2-deficient group was significantly lower than that in the similarly treated wt group and comparable to that in the control groups (Fig 1A). Evaluation of left ventricular cross-sectional area showed similar differences to the relative heart weight analysis, except that the values in AngII-treated wt animals were less elevated and therefore the comparison between the two AngII-treated groups did not reach the significance level (Fig 1B, 1G and 1H). The ventricular lumen area, as a measure of ventricular dilatation, was comparable in the two control groups and in the AngII-treated PAR2-deficient mice, and only in the AngII-treated wt mice was the ventricular lumen significantly increased by 1.7-fold compared with the AngII-treated PAR2-deficient animals (Fig 1C). In addition, we analysed the cardiac expression of different genes that are associated with cardiac hypertrophy including cardiac troponin (*Tnnt2*), *Fgf23* and natriuretic peptide type B (*Nppb*). Cardiac *Tnnt2* expression was comparable in controls but mean expression was about 2 times higher in AngII-treated wt mice. No increase in *Tnnt2* was seen in similar treated PAR2 deficient mice (Fig 1D). *Fgf23* expression was approximately 3-fold higher in AngII-treated wt mice compared to controls of both genotypes and nearly 2-fold higher compared to AngII-treated PAR2-deficient mice (Fig 1D). Again, because of the high variance within the AngII-treated groups, only a trend toward lower *Fgf23* expression could be observed in AngII-treated PAR2-deficient mice compared with AngII-treated wt mice (Fig 1E). In contrast, cardiac expression of the natriuretic peptide type B (*Nppb*) showed no significant differences between groups. However, highest mean values were also detected in AngII-treated wt mice (Fig 1F).

Finally, we investigated whether hypertrophy-promoting effects of PAR2 in our model are mediated by activation of MAP kinases. Interestingly, in the PAR2-deficient control mice, on average 4 times more cells were P-ERK1/2-positive than in the wt controls (Fig 2A). In the hearts of PAR2-deficient mice, P-ERK1/2-positive cells were found both in the endothelium of arterioles and in interstitial cells, especially in capillary endothelial cells (Fig 2B). In all other groups, P-ERK1/2-positive cells were significantly less frequent (Fig 2A and 2C), but in the same localization and mainly in endothelial cells of arterioles (Fig 2C). Since ERK1/2 phosphorylation also plays a role in fibrosis and numerous fibroblasts are P-ERK1/2-positive in our animal model, we have also examined this in isolated neonatal rat cardiac fibroblasts after PAR2 inhibition with AZ3541 with and without AngII stimulation. No significant differences were found between the groups studied, but the most pronounced phosphorylation of ERK1/2 was observed in unstimulated fibroblasts treated with PAR2 inhibitor (Figs 2D and S1), which is in line with the *in vivo* experiments with PAR2-deficient mice.

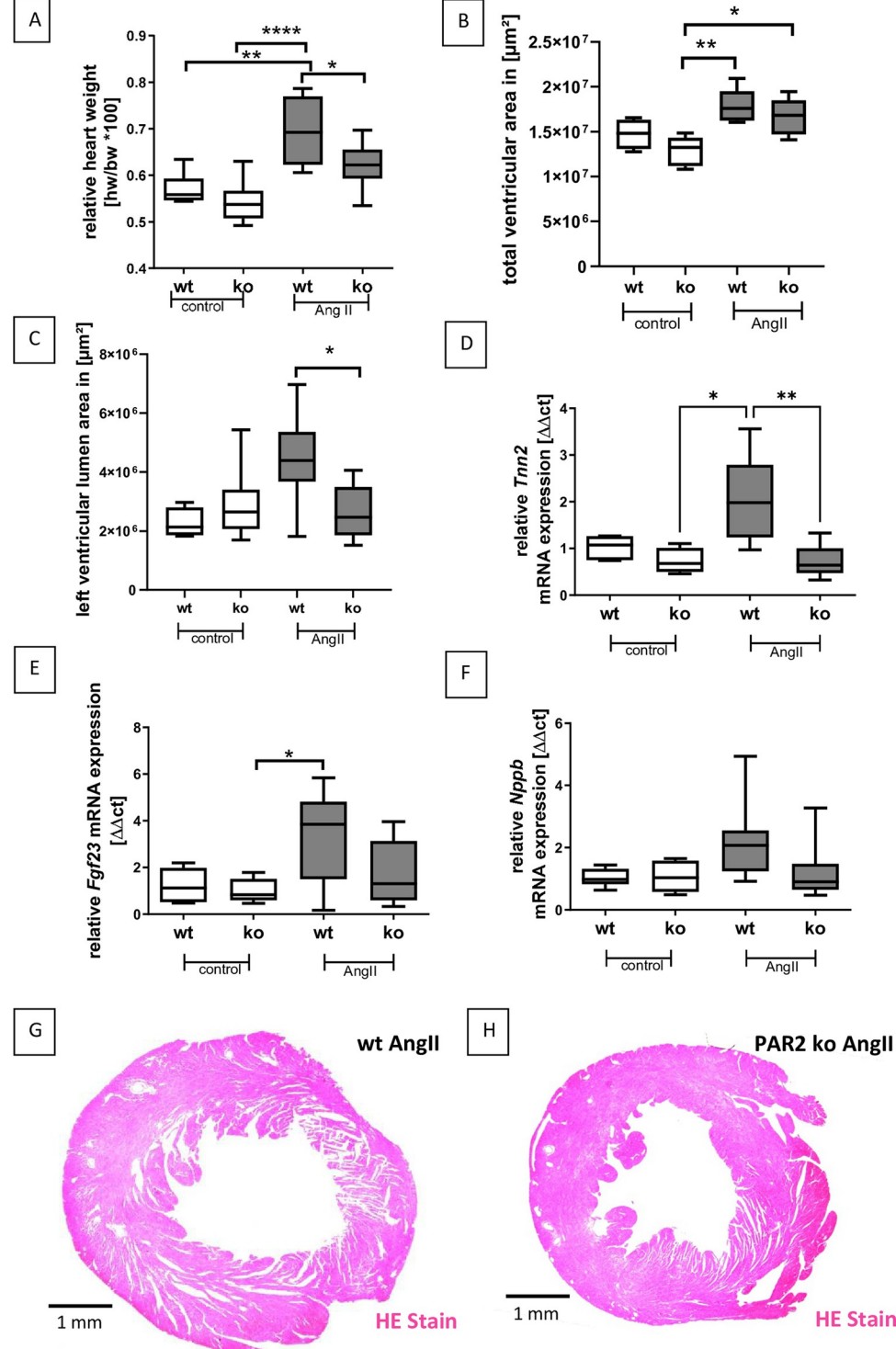

**Fig 1. AngII-induced LVH was reduced in PAR2 deficient mice. A:** Relative heart weight in relation to body weight is shown. **B:** Total cardiac section area was assessed on HE stain using CellSens software. **C:** Left ventricular lumen area was assessed using CellSens software. **D:** *Tnnt2* (cardiac Troponin) mRNA expression was measured using real-time PCR. **E:** *Fgf23* mRNA expression was measured using real-time PCR. **F:** *Nppb* (natriuretic peptide type B) mRNA expression was measured using real-time PCR. **G:** A representative hematoxilin/eosin (HE) stained heart section of a wt AngII-treated mouse is shown. **H:** A representative hematoxilin/eosin (HE) stained heart section of a PAR2-deficient AngII-treated mouse is shown. Wt control (n = 6) and PAR2 deficient controls (n = 6) were compared

with AngII-treated wt (n = 8) and PAR2 deficient mice (n = 9). Significant differences were marked: *p<0.05; **p<0.01; ****p<0.0001.

## 3.2 Wall thickening of intramycardial arterioles and hypertrophy of the left ventricle are differentially affected by PAR2 deficiency in AngII-treated mice

Next, we focussed on PAR2-mediated vascular changes in the AngII mouse model. In healthy untreated mice, arteriolar wall thickness was comparable in both genotypes with a mean thickness ranging between 8.9 μm and 9.2 μm and increased 1.3-fold in wt mice after AngII-treatment, but did not reach the level of significance (Fig 3A, 3C and 3E). While PAR2 deficient mice showed reduced LVH after AngII treatment compared with wt animals, arteriolar wall thickness significantly increased 1.8-fold compared to healthy control groups and AngII-treated wt group (Fig 3A, 3D and 3F). An analysis of cardiac arterioles with a wall thickness >18μm showed that these were significantly more frequent in AngII-treated PAR2-deficient mice than in the same treated wt animals; no vessels with a wall thickness >18μm could be detected in the untreated controls (Fig 3B).

## 3.3 Myocardial capillary supply tended to be reduced in PAR2-deficient mice and after treatment with AngII

To investigate the role of PAR2 in vascularization of the hypertrophic heart, we stained for CD31 as a marker for endothelial cells and analysed regions in which the cardiomyocytes were cut transversely (Fig 4A). Here, the number of capillaries per area showed only minor differences. The number of capillaries per field of vision tended to be highest in healthy wt mice (14.8±2.1) to the lowest in PAR2-deficient mice treated with AngII (11.94±1.9) (Fig 4B). Although no significance could be demonstrated, the mean values for the wt groups were higher compared to the corresponding PAR2-deficient group receiving the same treatment (Fig 4B).

## 3.4 PAR2 is involved in regulating cardiac fibrosis in the AngII hypertension model

To study fibrosis in PAR2-deficient mice, we examined total collagen content with Sirius-Red and collagen IV and α-SMA by immunohistochemistry. In control animals of both genotypes, total collagen, measured as the percentage of Sirius Red positive area in transversal heart sections, was low and at comparable levels, ranging between 0.37–0.68% of the section area (Fig 5A). In AngII-treated wt mice, this percentage of Sirius-Red positive area was on average approximately 3-fold higher on average, but showed a high variance such that fibrosis did not reach the significance level compared with control animals (Fig 5A and 5D). In contrast, total collagen was increased 10- and 20-fold in AngII-treated PAR2-deficient animals compared to wt and PAR2-deficient control animals and also significantly elevated compared to AngII treated wt mice (Fig 5A and 5E). The results of the Sirius Red-based determination of total collagen were essentially confirmed by the immunohistochemical evaluation of cardiac collagen IV. Again, the mean score was below 2 in both control groups and showed a tendency toward lower values in the PAR2-deficient animals (Fig 5B). Collagen IV was only slightly increased in the AngII-treated wt mice compared to controls but reached the significance level (Fig 5B). Consistent with the results of Sirius Red evaluation, the collagen IV score was significantly increased in AngII-treated PAR2 deficient mice compared with controls of both genotypes and also AngII-treated wt mice (Fig 5B). However, quantification of collagen I mRNA revealed

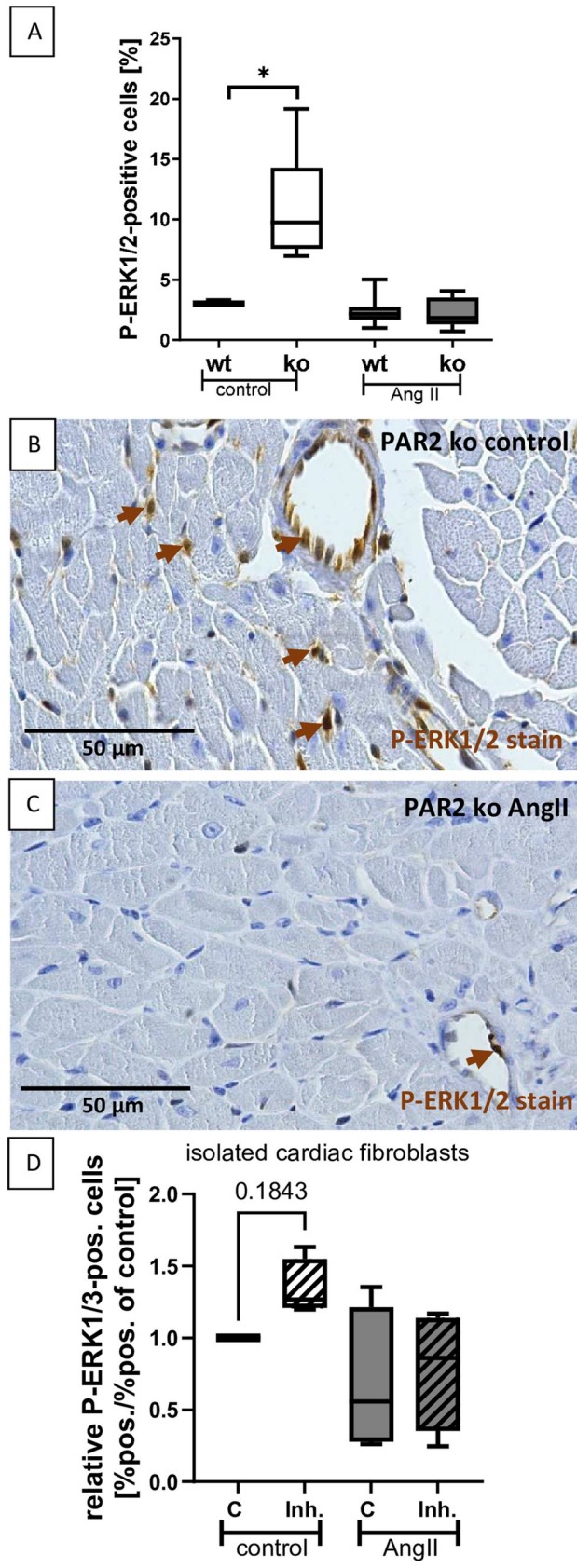

**Fig 2. Cardiac signaling via ERK1/2 was increased in untreated PAR2 deficient mice. A:** The percentage of P-ERK1/2 positive cells in mouse heart sections was evaluated using Qupath software. **B:** A representative picture of a heart section stained for P-ERK1/2 using IHC of a PAR2 deficient control mouse is shown. Examples of P-ERK1/2-positive cells are marked by brown arrows. **C:** A representative picture of a heart section stained for P-ERK1/2 using IHC of a PAR2 deficient AngII-treated mouse is shown. Wt control (n = 6) and PAR2 deficient controls (n = 6) were compared with AngII-treated wt (n = 8) and PAR2 deficient mice (n = 9). Significant differences were marked: *p<0.05. **D:** The effect of PAR2 inhibition using 2 μM AZ3541 on relative ERK1/2 phosphorylation in isolated rat cardiac fibroblasts in unstimulated controls and AngII-stimulated cells was shown, as assessed by immunofluorescence staining (4 replicates per group were used).

no significant differences between the four groups (S2 Fig). In addition, staining for alpha-SMA, a marker of myofibroblasts, was low and restricted to vascular smooth muscle cells (VSMC) in control mice of both genotypes and AngII-treated wt animals (Fig 5C and 5F), but slightly increased in AngII-treated PAR2-deficient mice (Fig 5C and 5G). To investigate the role of the major profibrotic cytokine TGF-β in PAR2 regulated cardiac fibrosis we analysed the phosphorylation of the TGF-β downstream signalling molecule Smad2/3. The percentage of P-Smad2/3-positive cells in cardiac sections showed relatively high variation particularly in AngII-treated wt mice (Fig 5H). Most P-Smad2/3-positive cells were interstitial cells (Fig 5I). In PAR2 deficient mice treated with AngII, the percentage of P-Smad2/3-positive cells was lowest, but did not reach significance level to other groups (Fig 5H), suggesting that the pro-fibrotic effects of PAR2 deficiency are presumably TGF-β independent. Smad3 phosphoryla-tion experiments were also performed on isolated neonatal rat cardiac fibroblasts treated either with the PAR2 inhibitor AZ3541 and simultaneously with AngII or with AZ3541 without AngII stimulation. Treatment with the PAR2 inhibitor showed no significant effect on the phosphorylation of the TGF-ß signalling molecule Smad3. There was only a trend towards increased Smad3 phosphorylation in fibroblasts when stimulated with AngII alone (Figs 5J and S3).

To further investigate the PAR2-mediated effects on cardiac fibrosis, gene expression analy-ses were also performed in neonatal rat cardiac fibroblasts. In addition to the above mentioned PAR2 inhibitor AZ3541, a PAR2 blocking antibody (SAM11) was also used. Moreover, in addition to untreated fibroblasts, fibroblasts treated with the isotype of the antibody served as a further control. As expected, the relative mRNA expression of the myofibroblast markers SMA (*Acta2*, Fig 6A, only a tendency), collagen 1 (Fig 6B) and collagen 4 (Fig 6C) was upregu-lated after AngII-stimulation, while AngII had no significant effect on fibronectin (Fig 6D) and TGFß-expression (Fig 6E) in fibroblasts. However, both PAR2 inhibitors significantly increased the mRNA expression of collagen 1, fibronectin and TGF-ß in unstimulated fibro-blasts, but not in AngII-stimulated cells. In summary, in contrast to the mouse model, PAR2 inhibition in AngII-stimulated fibroblasts tended to decrease the expression of SMA, collagen 1, and collagen 4 (Fig 6A–6C). This is in contrast to the mouse model in which AngII treat-ment caused increased fibrosis in PAR2-deficient mice (Fig 5A–5G).

## 3.5 PAR2 is barely expressed in cardiac myocytes, but stronger in interstitial and vascular smooth muscle cells

Heart mRNA expression analysis confirmed that the PAR2 receptor was not expressed in the PAR2-deficient mice above background (Fig 7A). In the wt mice, PAR2 receptor expression was clearly detectable and significantly decreased after AngII treatment, PAR2 receptor expression was significantly reduced by approximately 40% in the wt mice (Fig 7A). After fail-ing to find a suitable antibody to localize the PAR2 receptor in mouse hearts by immunohis-tochemistry, we performed *in situ* hybridization for PAR2 to investigate which cardiac cells

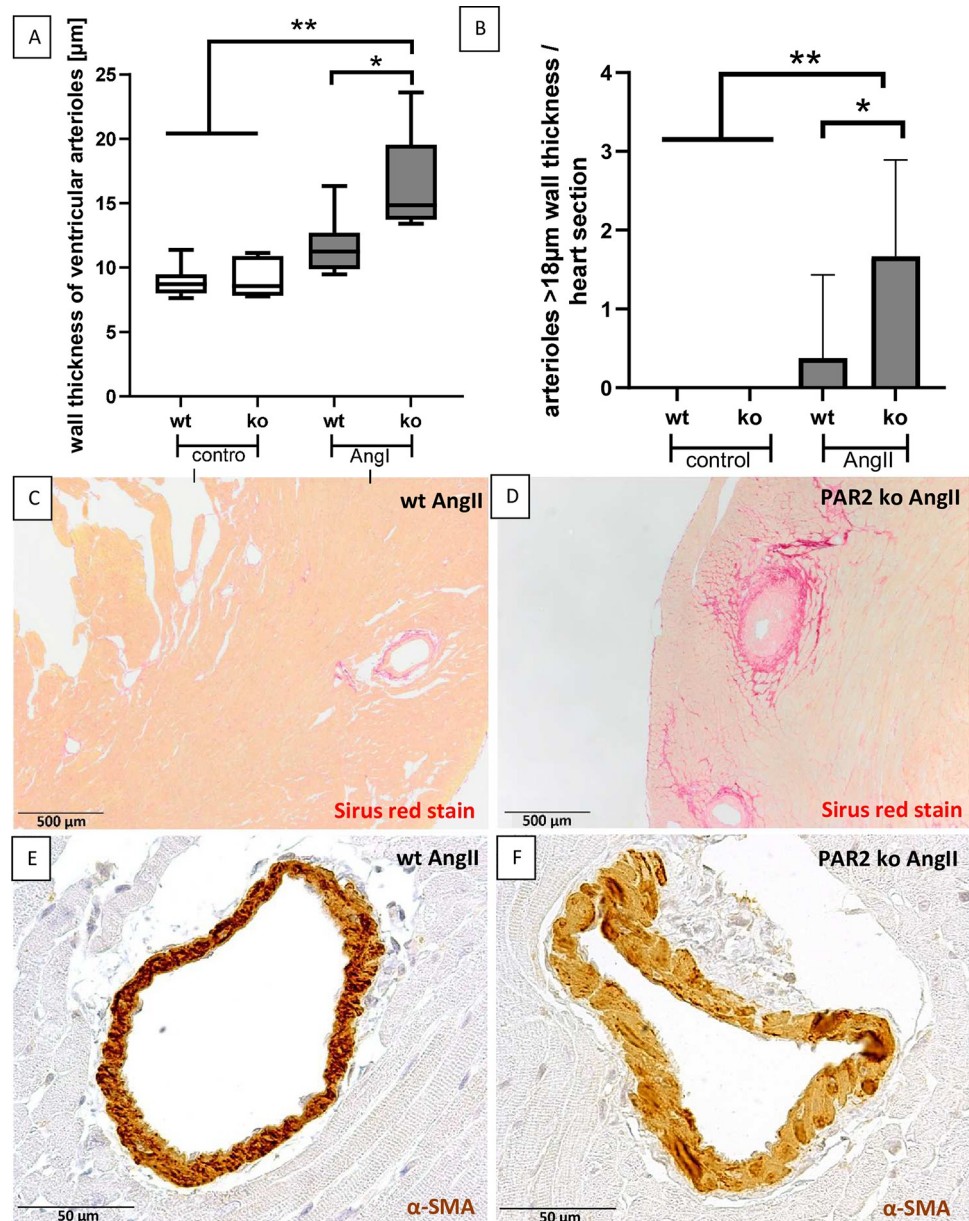

**Fig 3. Ventricular arteriolar wall thickness was increased in PAR2 deficient mice treated with AngII. A:** The average wall thickness of intramural cardiac arterioles in the left ventricle was evaluated. **B:** The number of arterioles with a wall thickness $\geq$ 18 μm per heart section is shown. **C:** Overview of a Sirius Red stained left ventricular section showing representative arterioles in wt mice treated with AngII. **D:** Overview of a Sirius Red stained left ventricular section showing representative arterioles in PAR2 deficient mice treated with AngII **E:** Detail of a representative arteriole in a wt AngII-treated mouse stained for α-smooth muscle actin (SMA). **F:** Detail of a representative arteriole in a PAR2 deficient AngII-treated mouse stained for α-smooth muscle actin (SMA). Wt control (n = 6) and PAR2 deficient controls (n = 6) were compared with AngII-treated wt (n = 8) and PAR2 deficient mice (n = 9). Significant differences were marked: *p<0.05; **p<0.01.

express this receptor and are thus potentially involved in the regulation of PAR2-regulated processes including hypertrophy and fibrosis in the AngII mouse model.

As expected, *in situ* hybridization for PAR2 was negative in cardiac tissue of the PAR2 deficient animals (Fig 7B). In cardiac tissue of wt animals PAR2 expression was not restricted to a

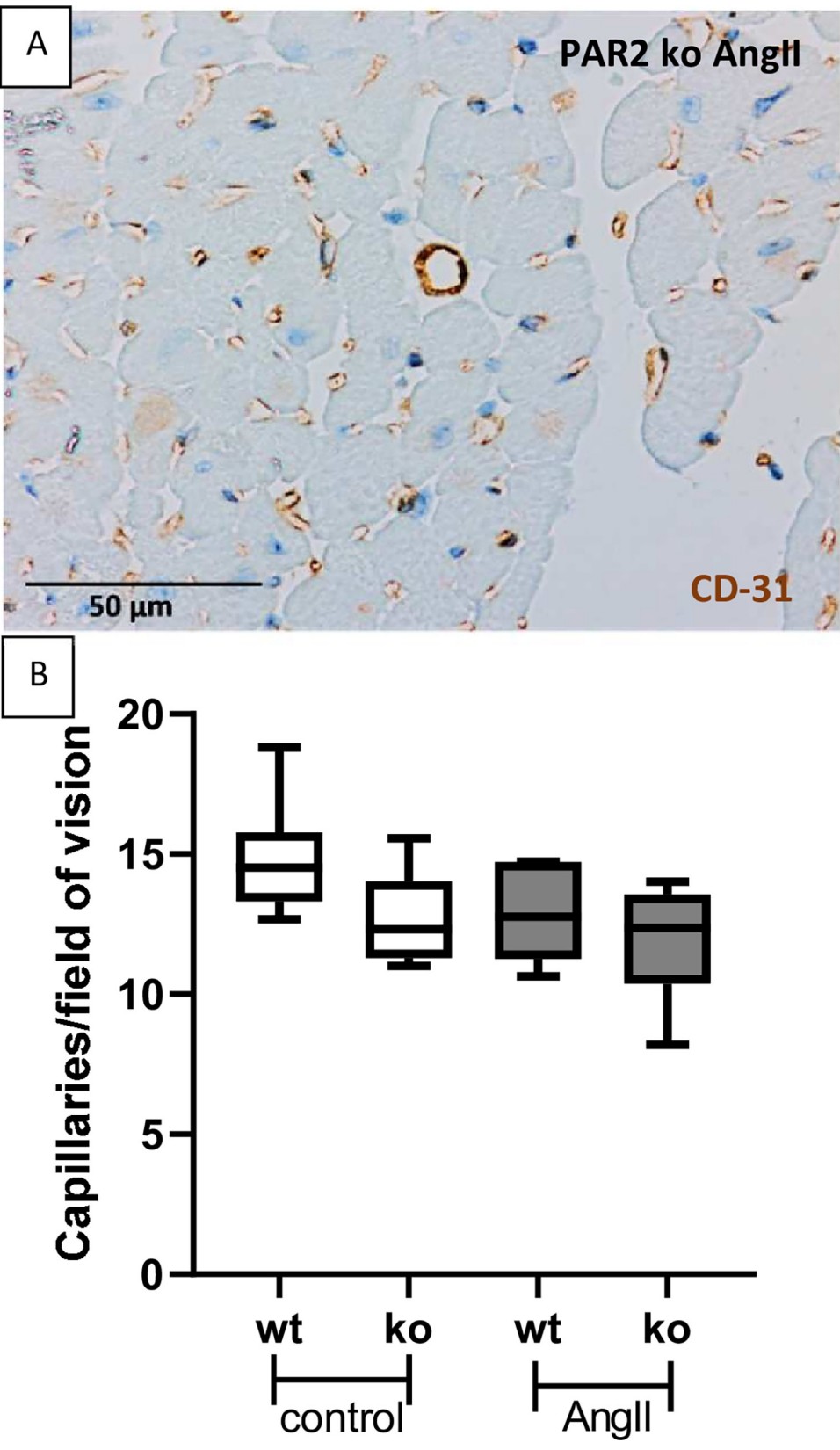

**Fig 4. Capillary supply of myocard and vascularization was not significantly affected by PAR2. A:** CD31 stained by IHC was used to analyze of capillary density in transverse cut cardiac muscle tissue as demonstrated in an untreated wt control mouse. **B:** Mean capillary densities was shown in transverse cut cardiac muscle tissue in control and AngII-treated sections. Wt control (n = 6) and PAR2 deficient controls (n = 6) were compared with AngII-treated wt (n = 8) and PAR2 deficient mice (n = 9).

single cell type in cardiac tissue. We detected relatively strong PAR2 expression in VSMC (Fig 7C and 7E, red arrows) while endothelial cells (Fig 7C, cyan arrow), interstitial cells or inflammatory cells, showed comparatively low PAR2 expression (Fig 7C and 7E, green arrows). In cardiomyocytes PAR2 expression was also low, as demonstrated by low numbers of PAR2-positive dots (Fig 7D, black arrows).

# 4. Discussion and conclusion

## 4.1. Discussion

In this study, we used PAR2-deficient mice to explore the role of PAR2 in cardiac changes after AngII infusion, a hypertension model. AngII is known to affect cardiac function through contractility, hypertrophy and fibrosis via the AT-1 receptor [25].

**4.1.1. LVH and PAR2.** Our first finding showed reduced cardiac hypertrophy in PAR2 deficient mice, linked to lower levels of cardiac FGF23 and cardiac troponin. This aligns with other studies indicating PAR2's role in hypertrophy. In cultured neonatal rat cardiomyocytes, PAR2 induced hypertrophic growth in a MEK1/2 and p38-dependent manner, and cardiomyocyte-specific overexpression of PAR2 in mice induced heart hypertrophy [18,26]. Treatment of renin-overexpressing mice, as a model of hypertension, with the PAR2 antagonist FSLLRY for 4 weeks resulted in reduced left ventricular wall thickness [27]. However, the effect of PAR2 on hypertrophy depends on level of stimulation since low level stimulation of PAR2 resulted in anti-hypertrophic response [28]. Because we found reduced LVH in PAR2-deficient mice, AngII treatment appears to be more consistent with stronger stimulation. However, at the endpoint of our study we did not find reduced ERK1/2 activation in PAR2-deficient mice; instead, ERK1/2 phosphorylation was increased in healthy PAR2-deficient mice but equally low in both AngII-treated groups. It is likely that ERK1/2 phosphorylation is not permanently increased in the animal model because only a transient increase in ERK phosphorylation was detected in AngII-stimulated cardiomyocytes [26]. In our study, mainly interstitial cells such as fibroblasts and endothelial cells or possibly inflammatory cells are P-ERK1/2 positive. Our *in situ* hybridization analyses for PAR2 confirm that these cells also carry the PAR2 receptor. Therefore, we also performed cell culture experiments with primary cardiac fibroblasts. Although these experiments showed no significant differences between untreated and PAR2 inhibitor-treated cells, most P-ERK1/2-positive fibroblasts were again found in the samples in which PAR2 was inhibited and not stimulated with AngII. Next, we investigated TGF-β signalling as another potential pathway influenced by PAR2. AngII treatment increases the expression of the cytokine TGF-β, which can mediate hypertrophy in AngII-induced hypertrophy, as TGF-β-deficient mice exhibited reduced LVH compared to control animals [29]. One way PAR2 affects TGF-β signalling is by interacting with TGF-β receptor 1. Absence or downregulation of PAR2 was associated with downregulation of caveolin-1, leading to decreased internalization of pro-fibrotic PAR1 and TGF-β receptor [13]. Smad2/3 phosphorylation, as a marker of TGF-β activation, was variable without any significant differences between groups in the AngII animal experiment. Thus, these data do not help us to understand the cardiac changes in PAR2-deficient mice. However, studies of ERK1/2 and TGF-β pathways have always been performed in cell culture systems and are therefore

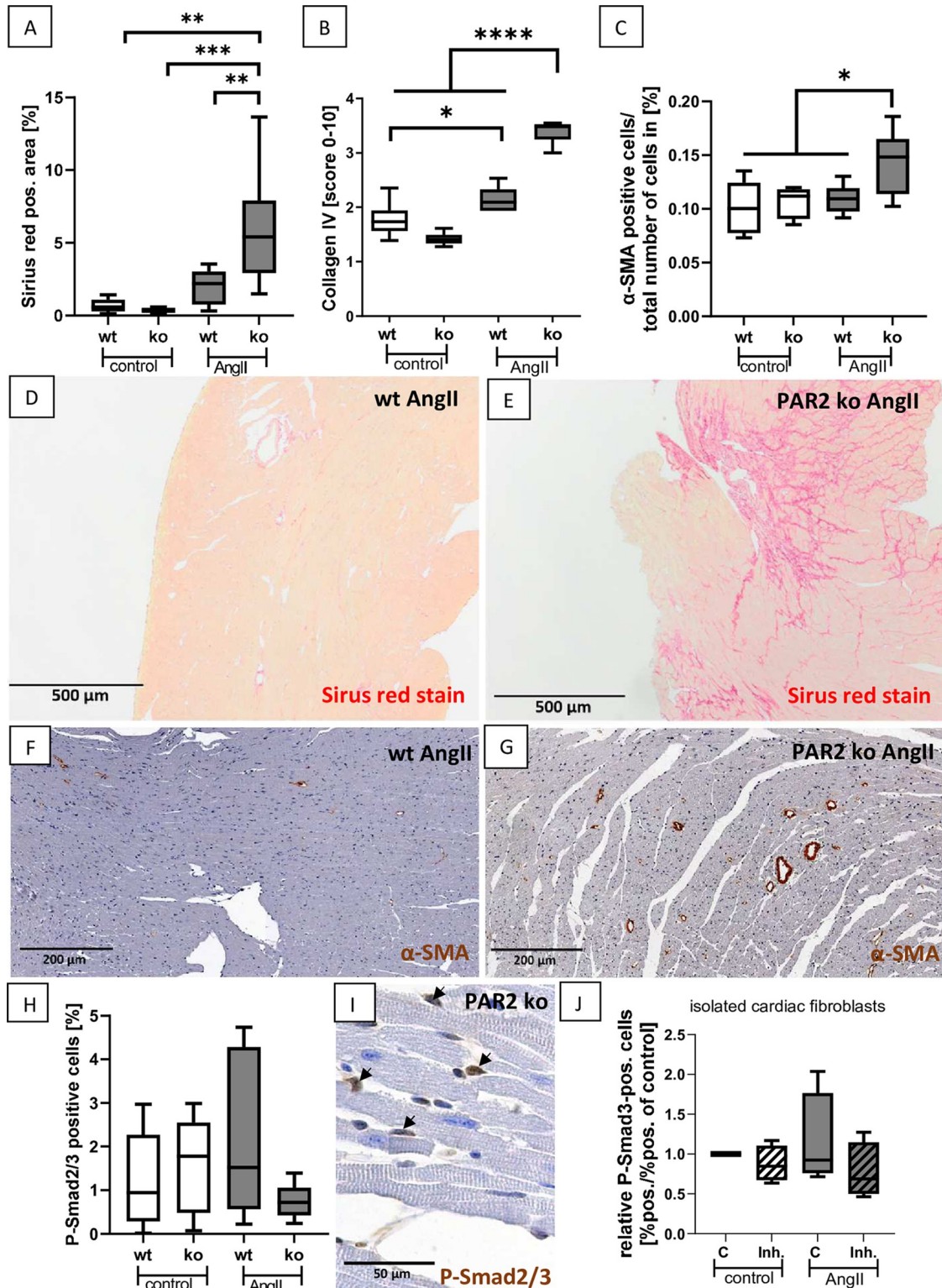

**Fig 5. Cardiac fibrosis is increased in AngII-treated PAR2 deficient mice. A:** Percentage of cardiac fibrosis, as assessed by Sirius Red staining, measured with MatLab software. **B:** Semi-quantitative score of cardiac collagen IV, as assessed by immunohistochemistry (IHC) staining. **C:** The percentage of cardiac myofibroblasts was evaluated as α-SMA positive cells/per total number of cells per cardiac section. **D:** A representative Sirius Red stained heart section of a wt AngII-treated mouse is shown. **E:** A representative Sirius Red stained heart section of a PAR2 deficient AngII treated mouse is shown. **F:** A representative heart section

stained for α-SMA using IHC of a wt AngII-treated mouse is shown. **G:** A representative heart section stained for α-SMA using IHC of a PAR2 deficient AngII-treated mouse is shown. **H:** Computer-assisted evaluation of the percentage of cardiac P-Smad2/3-positive cells. **I:** A representative picture of a heart section from a PAR2 deficient control mouse stained by IHC (marked by arrows) is shown. Wt control (n = 6) and PAR2 deficient controls (n = 6) were compared with AngII-treated wt (n = 8) and PAR2 deficient mice (n = 9). Significant differences were marked: *p<0.05; **p<0.01; ***p<0.001; ****p<0.0001. **J:** D: The effect of PAR2 inhibition using 2 μM AZ3541 on relative Smad3 phosphorylation in isolated rat cardiac fibroblasts in unstimulated controls and AngII-stimulated cells was shown, as assessed by immunofluorescence staining (4 replicates per group were used).

difficult to translate to a complex, long-term animal experiment. However, our own experiments *in vitro* using primary cardiac fibroblasts showed no significant differences between groups. In contrast, *Tgfb2* mRNA expression was significantly increased by both PAR2 inhibitors used, but only in unstimulated fibroblasts. Interestingly, in PAR2-deficient mice, cardiac vascular smooth muscle cells responded to AngII stimulation with hypertrophy, in contrast to cardiomyocytes. An important role of PAR2 for VSMC is supported by our observation that VSMC express PAR2 at higher levels than cardiomyocytes. However, previous studies in which PAR2 was overexpressed in VSMC and led to proliferation of these cells also suggested a pro-hypertrophic effect for this cell type [30]. Since vascular hypertrophy is again mediated by TGF-β, PAR2-induced TGF-β receptor internalization might explain the unexpected findings.

**4.1.2. Cardiac fibrosis and PAR2.** The second finding in our study was, that PAR2 deficiency enhanced cardiac fibrosis after treatment with AngII. Staining for collagen IV, Sirius Red, and alpha-SMA revealed significantly greater left ventricular fibrosis in PAR2-/- mice compared with wt mice after treatment with AngII, suggesting an anti-fibrotic effect of the PAR2 receptor *in vivo*. In our *in vitro* experiments using cardiac fibroblasts we observed significantly increased expression of the TGF-ß target genes *Col1a1* and *Fn1* in the unstimulated cells which was in line with the increased TGF-ß upregulation under these conditions. However, in AngII-stimulated fibroblasts, the PAR2 inhibitor did not further increase the expression of these matrix genes, but rather tended to decrease their expression. The role of PAR2 in the development of fibrosis is still unclear, as both pro- and anti-fibrotic effects have been described. However, PAR2 has shown pro-fibrotic effects in most studies. There are numerous studies showing that cardiac hypertrophy is usually associated with fibroblast activation and fibrosis [31,32], and that both hypertrophy and fibrosis can be reduced by, for example, RAAS blockade [33]. It is therefore unusual that these two processes appear to be separate in our study. In most cases, the two processes of cardiac hypertrophy and fibrosis are coupled, i.e. a reduction in hypertrophy is usually associated with a reduction in fibrosis. However, inhibition of hypertrophy alone may also lead to adverse cardiac effects in the setting of sustained volume overload [34]. If the absence of PAR2 only affects hypertrophy, an activation of fibroblasts would be conceivable, as previous studies in this model with PAR2-deficient mice have shown that blood pressure is not significantly reduced even in PAR2-deficient mice [35]. This means that the fibroblasts may continue to be stimulated by the elevated blood pressure. In contrast to our observation that cardiac fibrosis was significantly higher in PAR2-deficient mice than in wt animals when exposed to the same treatment, the study by Matsuura et al. showed exactly the opposite [36]. Here, significant interstitial fibrosis occurred in the left atrium of wt mice as early as 2 weeks after AngII infusion, which was significantly reduced in PAR2-deficient mice receiving the same treatment [36]. It is unclear why we obtained such contrasting results in the same AngII-induced animal model. As in our experiment, cardiac PAR2 expression after AngII treatment tended to be lower than in healthy controls [36]. However, the experiment was terminated earlier and we examined fibrosis in the left ventricle rather than the left atrium. Data obtained by the same research group in spontaneously

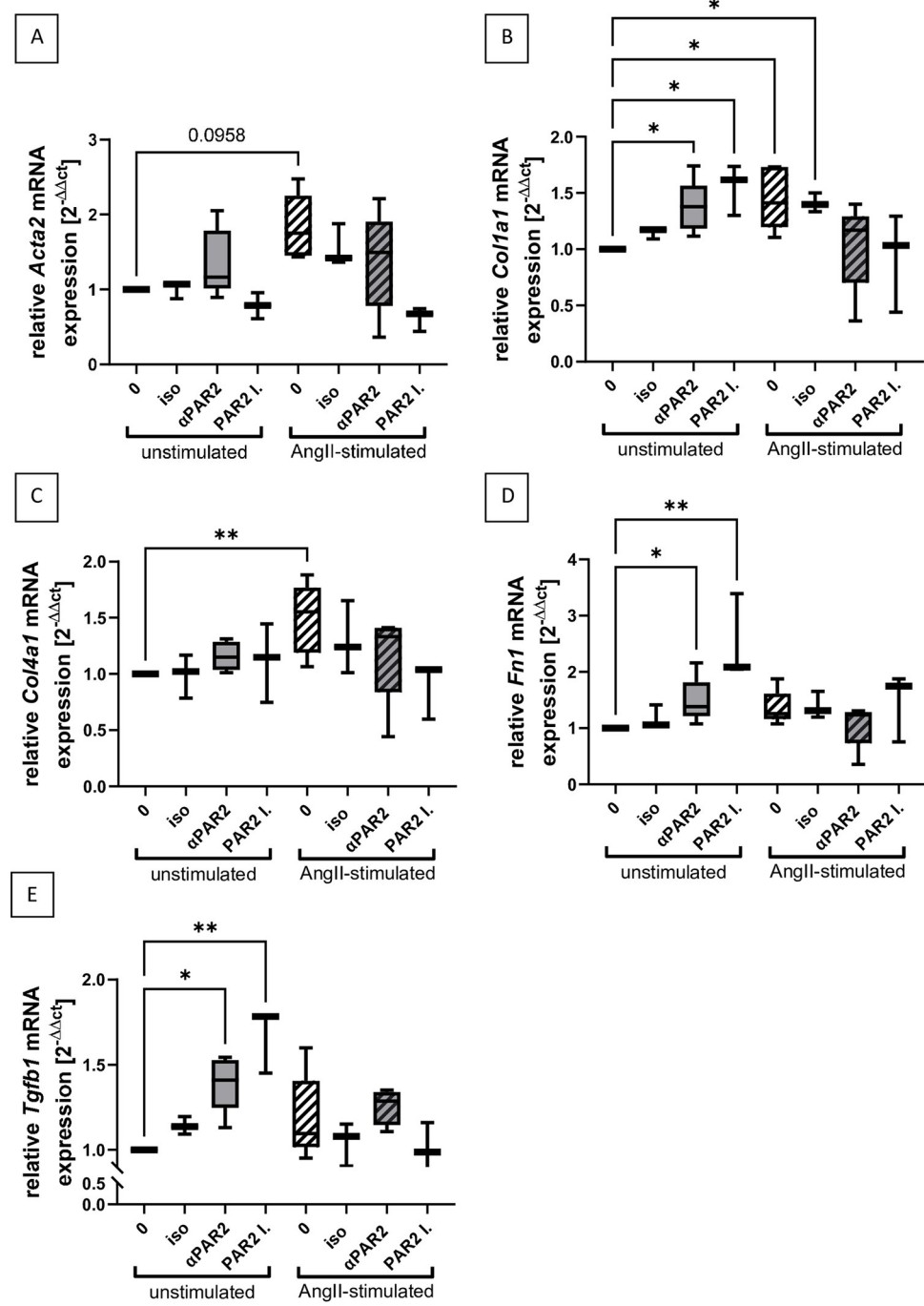

**Fig 6. Changes in myofibroblast markers and matrix molecule mRNA expression in isolated cardiac fibroblasts after AngII stimulation and PAR2 inhibition.** Cardiac fibroblasts isolated from neonatal rats were treated with anti-PAR2 blocking antibody SAM11 (αPAR2, n = 5 each) and 2μM PAR2 inhibitor AZ3541 (PAR2 I., n = 3 each) or left untreated (0, n = 5 each) or treated with isotype control antibody (iso, n = 3) as a control using unstimulated or AngII stimulated cells (hatched bars). After lysis of fibroblasts mRNA levels of myofibroblast markers **A:** *Acta2* (SMA), **B:** *Col1a1* (collagen 1), matrix molecules **C:** *Col4a1* (collagen 4) and **D:** *Fn1* (fibronectin) and the pro-fibrotic cytokine **E:** *Tgfb1* (TGF-ß) were measured. Significant changes compared to untreated control (0) were marked: *p<0.05; **p<0.01.

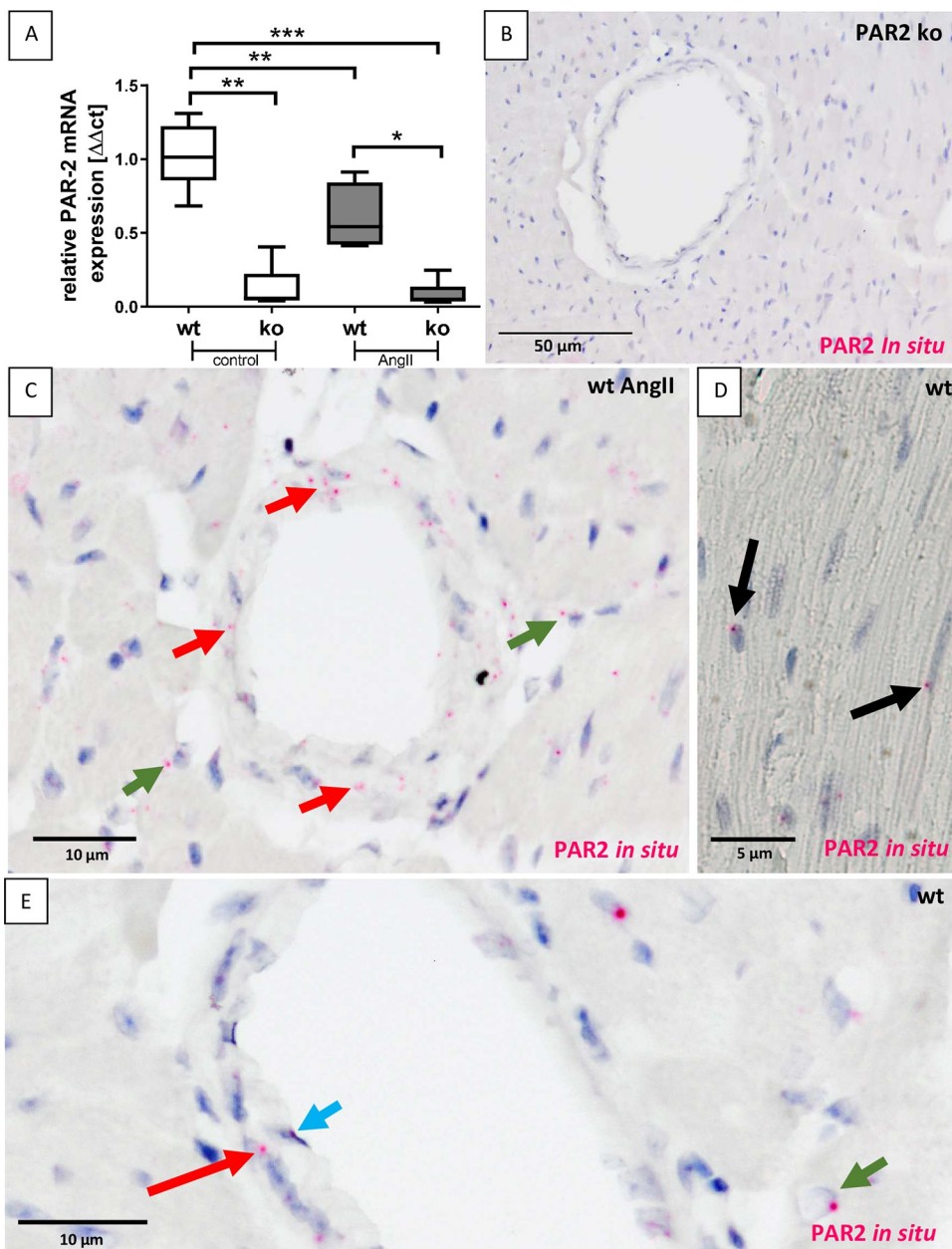

**Fig 7. Expression and localization of PAR2 in mouse hearts. A:** The lack of PAR2 in PAR2 deficient mice was confirmed by real-time PCR using cardiac tissue. Wt control (n = 6) and PAR2 deficient controls (n = 6) were compared with AngII-treated wt (n = 8) and PAR2 deficient mice (n = 9). Significant differences were marked: *p<0.05; **p<0.01; ***p<0.001. **B:** No staining for PAR2 *in situ* hybridization was found in cardiac sections of a PAR2 deficient mouse. **C:** In cardiac sections of wt AngII-treated mice PAR2 expression was visualized using *in situ* hybridization. PAR2 expression, as shown by purple dots is seen in vascular smooth muscle cells (red arrows) and interstitial cells (green arrows). **D:** Low PAR2 expression is seen in cardiomyocytes (black arrows). **E:** PAR2 Expression is shown in vascular smooth muscle cells (red arrow), endothelial cells (cyan arrow) and interstitial cells suspicious for inflammatory cells (green arrow).

hypertensive rats support the pro-fibrotic effect of PAR2 in hypertensive heart injury. Spontaneously hypertensive rats treated with rivaroxaban, which inhibits activated factor X and may therefore reduce the activation of PAR2, also developed less cardiac fibrosis. When PAR2 was

inhibited with the PAR2 antagonist FSLLRY in the hypertensive renin transgenic mouse model collagen-3 expression was reduced [27], also supporting the pro-fibrotic effect of PAR2 in hypertension in the heart. It has been shown that PAR2 is expressed on fibroblasts cultured *in vitro* and that activation of the receptor leads to increased proliferation of these cells [16,37] and induces a pro-fibrotic phenotype in atrial fibroblasts expressing increased collagen and fibronectin [37]. Pro-fibrotic effects of PAR2 were also described in different models of kidney diseases [38] and injury models in other organs [39,40]. In contrast, anti-fibrotic effects of PAR2 are rarely found. For example, PAR2 deficient mice developed increased fibrosis as shown by excessive glomerular collagen deposition in a model of diabetic nephropathy [41]. Furthermore, PAR2 appears to protect against age-related cardiac fibrosis. Endomyocardial biopsies from elderly patients with heart failure and preserved ejection fraction showed that reduced cardiac PAR2 expression was associated with increased myocardial fibrosis [13]. These findings were confirmed in 1-year-old PAR2-deficient mice, which also showed increased age-dependent alpha-smooth muscle actin expression and collagen 1 deposition compared to wt controls [13].

Our study is constrained by the limited ability to investigate PAR2-mediated signalling. The regulation of the PAR2 receptor and its influences in the heart appear to be diverse and partly contradictory. Due to the large number of different activating proteases, receptor-receptor interactions and feedback loops, a variety of effects of the PAR2 receptor are already known.

Since the activation of the receptor results in various, even opposing effects, a precise understanding of the regulation is of crucial importance. Since some of the results in our study showed substantial alterations in cardiac tissue after a very short period of time, the PAR2 receptor may be a potent target for influencing cardiac remodelling with pharmaceuticals. Further studies are needed to elucidate the exact interaction of PAR2 in AngII-induced cardiac injury.

### 4.2 Conclusion

PAR2 plays an important role in the regulation of cardiac remodelling. We could demonstrate that PAR2 deficiency effectively protects against AngII-induced myocardial hypertrophy. In contrast, PAR2 deficiency enhances cardiac fibrosis and pathological vascular changes in the AngII mouse model, while inhibition of PAR2 in cardiac fibroblasts resulted only in untreated cells in significant increase of matrix genes. Thus, PAR2 has multiple effects in AngII-stimulated cardiac tissue and its role in cardiac remodelling appears to be ambivalent and dependent on the context.

### Supporting information

**S1 Table. Primer used for real-time PCR.**
(DOCX)

**S1 File. Detailed parameters used for the evaluation of P-ERK1/2 and P-Smad3-positive cardiac rat fibroblasts.**
(DOCX)

**S1 Fig. ERK1/2 phosphorylation in rat neonatal cardiac fibroblasts after inhibition of PAR2 und stimulation with angiotensin II.**
(TIF)

**S2 Fig. Relative Expression of Col1a1 mRNA in hearts from healthy and AngII-treated wt and PAR2 deficient mice.**
(TIF)

**S3 Fig. Smad3 phosphorylation in rat neonatal cardiac fibroblasts after inhibition of PAR2 und stimulation with angiotensin II.**
(TIF)

## Acknowledgments

The present work was performed in (partial) fulfillment of the requirements for obtaining the degree 'Dr. med.' from the Friedrich-Alexander University of Erlangen-Nürnberg for Albrecht Meyer zu Schwabedissen. The technical assistance of Anne Diener, Christina Mayer, Stefan Söllner, Miriam Reutelshöfer and Tajana Ries is gratefully acknowledged.

## Author Contributions

**Conceptualization:** Kerstin Amann, Felix B. Engel, Christoph Daniel.

**Data curation:** Silvia Vergarajauregui, Felix B. Engel.

**Formal analysis:** Albrecht Meyer zu Schwabedissen.

**Funding acquisition:** Christoph Daniel.

**Investigation:** Albrecht Meyer zu Schwabedissen, Silvia Vergarajauregui.

**Methodology:** Albrecht Meyer zu Schwabedissen, Silvia Vergarajauregui, Felix B. Engel, Christoph Daniel.

**Project administration:** Christoph Daniel.

**Resources:** Marko Bertog, Felix B. Engel.

**Supervision:** Christoph Daniel.

**Validation:** Silvia Vergarajauregui, Christoph Daniel.

**Writing – original draft:** Albrecht Meyer zu Schwabedissen, Christoph Daniel.

**Writing – review & editing:** Albrecht Meyer zu Schwabedissen, Silvia Vergarajauregui, Marko Bertog, Kerstin Amann, Felix B. Engel, Christoph Daniel.

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
