## [Decision Letter · Decision Letter 0]

25 Apr 2024

PONE-D-24-00413Protease-activated receptor 2 deficient mice develop less angiotensin II induced left ventricular hypertrophy but more cardiac fibrosisPLOS ONE

Dear Dr. Daniel,

Thank you for submitting your manuscript to PLOS ONE. After careful consideration, we feel that it has merit but does not fully meet PLOS ONE’s publication criteria as it currently stands. Therefore, we invite you to submit a revised version of the manuscript that addresses the points raised during the review process.

We look forward to receiving your revised manuscript.

Kind regards,

Luis Eduardo M Quintas, Ph.D.

Academic Editor

PLOS ONE

Reviewers' comments:

Reviewer's Responses to Questions

**Comments to the Author**

1. Is the manuscript technically sound, and do the data support the conclusions?

Reviewer #1: Yes

Reviewer #2: Partly

2. Has the statistical analysis been performed appropriately and rigorously? 

Reviewer #1: Yes

Reviewer #2: Yes

3. Have the authors made all data underlying the findings in their manuscript fully available?

Reviewer #1: Yes

Reviewer #2: Yes

4. Is the manuscript presented in an intelligible fashion and written in standard English?

Reviewer #1: Yes

Reviewer #2: Yes

5. Review Comments to the Author

Reviewer #1: Meyer zu Schwabedissen et. al conduct a study evaluating the effects of PAR2 deficiency on ang II-induced cardiac injury and find that it promotes fibrosis but reduces left ventricular hypertrophy.

The manuscript could be strengthened if the following points are addressed:

- What would explain the increase in p-ERK1/2 in the PAR2 KO healthy control animals? These results are drastically different than your other groups.

-Would sequencing the heart tissue provide insights into potential mechanisms regulating the enhanced fibrosis in the PAR2 KO Ang II groups?

- Given the variance it is hard to determine if there are true differences in P-Smad2/3, could this be measured by western blot instead? Same with the P-ERK1/2?

-Are blood pressure measurements, hemodynamics or echocardiography measurements possible to determine physiological and functional consequences of PAR2 deficiency?

Reviewer #2: Schwabedissen et al. investigated roles of protease activated receptor 2 (PAR2) on the heart using PAR2-KO mice. The mice were treated with angiotensin II (Ang II) for 4 weeks using osmotic minipumps, and then various in vivo studies were performed. They found that left ventricular hypertrophy (LVH) after Ang II treatment was inhibited in PAR2-KO mice, although cardiac interstitial fibrosis was significantly higher in PAR2-KO mice compared to wild-type mice. They concluded that PAR2 has multiple effects in AngII-stimulated cardiac tissue and its role in cardiac remodelling appears to be ambivalent. Although this paper is well-written and provides some scientific implications, there are severe concerns in the present study.

Major criticism:

1. Important findings in the present study is related to cardiac perivascular fibrosis. PAR2 deficiency with Ang II treatment enhanced cardiac perivascular fibrosis. However, the authors only showed historical data and their data are very descriptive without any mechanistic insights. At least some in vitro studies are required. Would it be possible to perform in vitro studies using isolated cardiac fibroblasts from PAR2-deficient mice?

2. The authors showed that cardiac hypertrophy induced by Ang II treatment was inhibited in PAR2-KO mice. How was this inhibition related with cardiac perivascular fibrosis? This is a critical question in this paper.

3. Although FGF23 mRNA expression levels in heart were shown in Figure 1, this is not enough. How were alpha-MHC and/or BNP mRNA expression levels in heart?

4. The authors showed that P-ERK1/2-positive cells in heart were significantly increased in PAR2-KO mice, but they were not increased in all other groups (Figure 2A). Why did these happen? Why did Ang II treatment decrease P-ERK1/2-positive cells in the heart frm PAR2-KO mice? More clear explanation and/or additional experiments were required to confirm this result.

5. The number of experiments or mice should be described in the main text and figure legends.

6. PLOS authors have the option to publish the peer review history of their article (what does this mean?). If published, this will include your full peer review and any attached files.

Reviewer #1: No

Reviewer #2: No

---

## [Author Response · Author response to Decision Letter 0]

6 Aug 2024

Reviewer #1: Meyer zu Schwabedissen et. al conduct a study evaluating the effects of PAR2 deficiency on ang II-induced cardiac injury and find that it promotes fibrosis but reduces left ventricular hypertrophy.

The manuscript could be strengthened if the following points are addressed:

- What would explain the increase in p-ERK1/2 in the PAR2 KO healthy control animals? These results are drastically different than your other groups.

This observation also surprised us the most. Other studies have shown that MAP kinases play an important role in the regulation of hypertrophy and fibrosis. We therefore investigated this in our study. However, we did not expect that P-ERK1/2 will be increased in the healthy PAR2 deficient mice. Other studies have shown that activation of PAR2 leads to increased phosphorylation of ERK1/2, so it is surprising that we observed the opposite, increased activation in the healthy PAR2-deficient animals. Since ERK1/2 can also be activated by other pathways, the observed response could also be a kind of compensation. Such compensations can occur especially in knock-out animals. Our experiments with isolated fibroblasts, newly included in the revised manuscript, showed the strongest phosphorylation of ERK1/2 in the unstimulated fibroblasts treated with a PAR-2 inhibitor (see Figure 2D). However, these results reached not the level of significance.

-Would sequencing the heart tissue provide insights into potential mechanisms regulating the enhanced fibrosis in the PAR2 KO Ang II groups?

While it cannot be ruled out that sequencing of cardiac tissue may provide answers to the mechanism, we would hypothesize that it is primarily differences in signal transduction of stimuli that occur. These are very often not regulated at the level of gene regulation but at the level of post-translational modification, so that they would not be detected by gene sequencing. However, differences in signal transduction might influence expression of other genes.

- Given the variance it is hard to determine if there are true differences in P-Smad2/3, could this be measured by western blot instead? Same with the P-ERK1/2?

Unfortunately, we do not have any frozen tissue left from the experiment to carry out Western blot analyses for the phosphorylation of Smad 2/3 and ERK1/2. 

-Are blood pressure measurements, hemodynamics or echocardiography measurements possible to determine physiological and functional consequences of PAR2 deficiency?

In fact, in our experimental setting, we planned to include measurements of blood pressure. We started with intra-arterial measurements. However, we could not measure blood pressure in the PAR2ko mice because we could not fix the catheter in the carotid artery, as the arteries seem to be more fragile compared to the corresponding wild-type mice. After failing to obtain a measurement in 4 PAR2ko animals, we refrained from further measurements. However, there are published data examining blood pressure in wild-type and PAR2-deficient mice after stimulation with angiotensin II and reporting no significant differences between the two genotypes, only that some individuals of the PAR2-deficient mice tended to develop lower blood pressure values [1]. The fact that the observed effects in the PAR2-deficient patients are most likely not due to an altered blood pressure was added in the revised version of the manuscript with a reference to the publication by McGuire et al.

Reviewer #2: Schwabedissen et al. investigated roles of protease activated receptor 2 (PAR2) on the heart using PAR2-KO mice. The mice were treated with angiotensin II (Ang II) for 4 weeks using osmotic minipumps, and then various in vivo studies were performed. They found that left ventricular hypertrophy (LVH) after Ang II treatment was inhibited in PAR2-KO mice, although cardiac interstitial fibrosis was significantly higher in PAR2-KO mice compared to wild-type mice. They concluded that PAR2 has multiple effects in AngII-stimulated cardiac tissue and its role in cardiac remodelling appears to be ambivalent. Although this paper is well-written and provides some scientific implications, there are severe concerns in the present study.

Major criticism:

1. Important findings in the present study is related to cardiac perivascular fibrosis. PAR2 deficiency with Ang II treatment enhanced cardiac perivascular fibrosis. However, the authors only showed historical data and their data are very descriptive without any mechanistic insights. At least some in vitro studies are required. Would it be possible to perform in vitro studies using isolated cardiac fibroblasts from PAR2-deficient mice?

As suggested by the reviewer, we performed in vitro experiments with primary cardiac fibroblasts. We would have liked to perform these experiments with fibroblasts from PAR2-deficient mice, but at present we do not have homozygous PAR2-deficient animals available, as we regularly backcross the animals. Instead, with the help of colleagues, we have isolated cardiac fibroblasts from the hearts of neonatal rats and stimulated them with AngII. We investigated PAR2-dependent effects by adding a blocking anti-PAR2 antibody (SAM11) and a PAR2 inhibitor (AZ3451). To exclude the possibility that the addition of the antibody had non-specific effects, an isotype control was used. Gene expression analyses for TGF-ß and matrix molecules were performed as read-outs and signal transduction, in particular the phosphorylation of ERK1/2 and Smad 3, was investigated by immunofluorescence microscopy. Western blot analyses were not performed, as it was not possible to generate sufficient material for various analyses from the neonatal hearts available. 

To our surprise, the experiments with cardiac fibroblasts showed that the expression of TGF-ß and possibly as a consequence also of collagen 1 and fibronectin was upregulated by the addition of the two PAR2 inhibitors (Figure 6). However, this effect was not further enhanced in AngII-stimulated fibroblasts. The in vitro experiments show that in the short observation period of 24 h, which was chosen to capture the early signal transduction as far as possible, not all genes are increased in their expression by AngII, but only collagen 1 and collagen 4 and also SMA. Furthermore, the significant increase in expression that we observe in the unstimulated cells for TGF-ß, collagen 1 and fibronectin is not observed for SMA and collagen 4. This shows that not all matrix molecules are affected by PAR2 in the same way, or at least with different kinetics. Overall, the in vitro experiments confirmed that PAR2 deficiency or inhibition can also have pro-fibrotic effects. However, the mechanism is still unclear and needs to be clarified in future studies.

2. The authors showed that cardiac hypertrophy induced by Ang II treatment was inhibited in PAR2-KO mice. How was this inhibition related with cardiac perivascular fibrosis? This is a critical question in this paper.

There are numerous studies showing that cardiac hypertrophy is usually associated with fibroblast activation and fibrosis [1, 2], and that both hypertrophy and fibrosis can be reduced by, for example, RAAS blockade [3]. It is therefore unusual that these two processes appear to be separate in our study. In most cases, the two processes of cardiac hypertrophy and fibrosis are coupled, i.e. a reduction in hypertrophy is usually associated with a reduction in fibrosis. However, inhibition of hypertrophy alone may also lead to adverse cardiac effects in the setting of sustained volume overload [4]. If the absence of PAR2 only affects hypertrophy, an activation of fibroblasts would be conceivable, as previous studies in this model with PAR2-deficient mice have shown that blood pressure is not significantly reduced even in PAR2-deficient mice [5]. This means that the fibroblasts may continue to be stimulated by the elevated blood pressure. However, more data are needed to prove this hypothesis. These considerations were included in the discussion in the manuscript. 

3. Although FGF23 mRNA expression levels in heart were shown in Figure 1, this is not enough. How were alpha-MHC and/or BNP mRNA expression levels in heart?

As suggested by the reviewer, we examined other markers of hypertrophy including BNP and cardiac troponin expression in the mouse hearts. Indeed, cardiac troponin was significantly increased in wt mice treated with AngII and expression was normalized to control levels in PAR2-deficient mice (Figure 1D). A similar trend was observed in the expression analysis of BNP (Figure 1F). However, no significant differences were observed for BNP.

4. The authors showed that P-ERK1/2-positive cells in heart were significantly increased in PAR2-KO mice, but they were not increased in all other groups (Figure 2A). Why did these happen? Why did Ang II treatment decrease P-ERK1/2-positive cells in the heart from PAR2-KO mice? More clear explanation and/or additional experiments were required to confirm this result.

This observation was also completely unexpected to us, and we tried to address this phenomenon in the in vitro experiments as well. Interestingly, the experiments with the isolated fibroblasts mirrored the observations from the animal experiments. Also in cell culture, the highest phosphorylation was observed in the samples with PAR2 inhibitor without AngII stimulation, whereas after AngII administration rather fewer cells showed phosphorylation of ERK1/2 (Figure 2D). As mentioned above, more induction was also observed for other targets in the samples that were not stimulated with AngII (Figure 6). Unfortunately, we do not yet have a conclusive explanation for this observation.

5. The number of experiments or mice should be described in the main text and figure legends. 

As suggest, we included number of experiments and numbers of mice also in the legends.

[1] Liu Q, Li HY, Wang SJ, Huang SQ, Yue Y, Maihemuti A, Zhang Y, Huang L, Luo L, Feng KN, Wu ZK. Belumosudil, ROCK2-specific inhibitor, alleviates cardiac fibrosis by inhibiting cardiac fibroblasts activation. Am J Physiol Heart Circ Physiol. 2022;323:H235-h47.

[2] Chalise U, Hale TM. Fibroblasts under pressure: cardiac fibroblast responses to hypertension and antihypertensive therapies. Am J Physiol Heart Circ Physiol. 2024;326:H223-h37.

[3] Garvin AM, De Both MD, Talboom JS, Lindsey ML, Huentelman MJ, Hale TM. Transient ACE (Angiotensin-Converting Enzyme) Inhibition Suppresses Future Fibrogenic Capacity and Heterogeneity of Cardiac Fibroblast Subpopulations. Hypertension. 2021;77:904-18.

[4] Crozatier B, Ventura-Clapier R. Inhibition of hypertrophy, per se, may not be a good therapeutic strategy in ventricular pressure overload: other approaches could be more beneficial. Circulation. 2015;131:1448-57.

[5] McGuire JJ, Van Vliet BN, Halfyard SJ. Blood pressures, heart rate and locomotor activity during salt loading and angiotensin II infusion in protease-activated receptor 2 (PAR2) knockout mice. BMC Physiol. 2008;8:20.

---

## [Decision Letter · Decision Letter 1]

26 Aug 2024

Protease-activated receptor 2 deficient mice develop less angiotensin II induced left ventricular hypertrophy but more cardiac fibrosis

PONE-D-24-00413R1

Dear Dr. Daniel,

We’re pleased to inform you that your manuscript has been judged scientifically suitable for publication and will be formally accepted for publication once it meets all outstanding technical requirements.

Kind regards,

Luis Eduardo M Quintas, Ph.D.

Academic Editor

PLOS ONE

Additional Editor Comments (optional):

Reviewers' comments:

Reviewer's Responses to Questions

**Comments to the Author**

1. If the authors have adequately addressed your comments raised in a previous round of review and you feel that this manuscript is now acceptable for publication, you may indicate that here to bypass the “Comments to the Author” section, enter your conflict of interest statement in the “Confidential to Editor” section, and submit your "Accept" recommendation.

Reviewer #1: All comments have been addressed

Reviewer #2: All comments have been addressed

2. Is the manuscript technically sound, and do the data support the conclusions?

Reviewer #1: Yes

Reviewer #2: Yes

3. Has the statistical analysis been performed appropriately and rigorously? 

Reviewer #1: Yes

Reviewer #2: Yes

4. Have the authors made all data underlying the findings in their manuscript fully available?

Reviewer #1: Yes

Reviewer #2: Yes

5. Is the manuscript presented in an intelligible fashion and written in standard English?

Reviewer #1: Yes

Reviewer #2: Yes

6. Review Comments to the Author

Reviewer #1: (No Response)

Reviewer #2: The authors performed many additional experiments and responded very well. There is no further comment on this paper.

7. PLOS authors have the option to publish the peer review history of their article (what does this mean?). If published, this will include your full peer review and any attached files.

Reviewer #1: No

Reviewer #2: No

---

## [Editor Report · Acceptance letter]

29 Sep 2024

PONE-D-24-00413R1 

PLOS ONE

Dear Dr. Daniel, 

I'm pleased to inform you that your manuscript has been deemed suitable for publication in PLOS ONE. Congratulations! Your manuscript is now being handed over to our production team.

Kind regards, 

on behalf of

Dr. Luis Eduardo M Quintas 

Academic Editor

PLOS ONE